# DeepRetinotopy: Predicting the Functional Organization of Human Visual Cortex from Structural MRI Data using Geometric Deep Learning

**Fernanda L. Ribeiro**[1,2]                    FERNANDA.RIBEIRO@UQ.EDU.AU

[1] *School of Psychology, The University of Queensland; Brisbane QLD 4072; Australia*

[2] *Queensland Brain Institute, The University of Queensland; Brisbane QLD 4072; Australia*

**Steffen Bollmann**[3]                    STEFFEN.BOLLMANN@CAI.UQ.EDU.AU

[3] *Centre for Advanced Imaging, The University of Queensland; Brisbane QLD 4072; Australia*

**Alexander M. Puckett**[1,2]                    A.PUCKETT@UQ.EDU.AU

## Abstract

Whether it be in a man-made machine or a biological system, form and function are often directly related. In the latter, however, this particular relationship is often unclear due to the intricate nature of biology. Here we developed a geometric deep learning model capable of exploiting the actual structure of the cortex to learn the complex relationship between brain function and anatomy from structural and functional MRI data. Our model was not only able to predict the functional organization of human visual cortex from anatomical properties alone, but it was also able to predict nuanced variations across individuals.

**Keywords:** fMRI, retinotopy, visual hierarchy, cortical folding, manifold, surface model

## 1. Introduction

Over the past few years there has been an effort to generalize deep neural networks to non-Euclidean spaces such as surfaces and graphs – with these techniques collectively being referred to as geometric deep learning (Bronstein et al.). Here we demonstrate the power of these algorithms by using them to predict brain function from anatomy using MRI data. The visual hierarchy is comprised of a number of different cortical visual areas, nearly all of which are organized retinotopically. That is, the spatial organization of the retina is maintained and reflected in each of these cortical visual areas. This retinotopic mapping is known to be similar across individuals; however, considerable inter-subject variation does exist, and this variation has been shown to be directly related to variability in cortical folding patterns and other anatomical features (Benson and Winawer, 2018; Benson et al., 2014). It was our aim, therefore, to develop a neural network capable of learning the complex relationship between the functional organization of visual cortex and the underlying anatomy.

## 2. Methods

To build our geometric deep learning model, we used the open-source 7T MRI retinotopy dataset from the Human Connectome Project (Benson et al., 2018). This dataset includes 7T fMRI retinotopic mapping data of 181 participants along with their anatomical data represented on a cortical surface model. The data serving as input to our neural network

included curvature and myelin values as well as the connectivity among vertices forming the cortical surface and their spatial disposition. The output of the network was the retinotopic mapping value (i.e., polar angle or eccentricity) for each vertex of the cortical surface model.

Prior to model training, the 181 participants from the HCP dataset were randomly separated into three independent datasets: training (161 participants), development (10 participants), and test (10 participants) datasets. During training, the network learned the correspondence between the retinotopic maps and the anatomical features by exposing the network to each example in the training dataset. Model hyperparameters were then tuned by inspecting model performance using the development dataset. Finally, once the final model was selected, the network was tested by assessing the predicted maps for each individual in the test dataset (previously not seen by the network nor the researchers). Models were implemented using Python 3.7.3, Pytorch 1.2.0, and geometric PyTorch (Fey and Lenssen, 2019), a geometric deep learning extension of PyTorch. Training was performed using NVIDIA Tesla V100 Accelerator units.

Our final model included 12 spline-based convolution layers (Fey et al., 2018) (Figure 1), interleaved by batch normalization and dropout. We trained our model for 200 epochs with batch size of 1, learning rate at 0.01 for 100 epochs and that was then adjusted to 0.005, using Adam optimizer. Our models' learning objective was to reduce the difference between predicted retinotopic map and ground truth. This mapping objective was measured by the smooth L1 loss function.

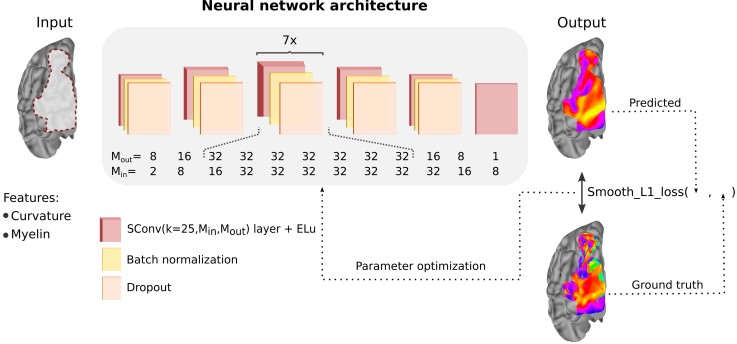

Figure 1: Geometric convolutional neural network architecture.

## 3. Results

Our neural network accurately predicted the main features of both polar angle and eccentricity retinotopic maps. Figure 2A shows the empirical (ground truth) and predicted polar angle and eccentricity maps of a single individual from the test dataset (Participant 1) for both left (LH) and right (RH) hemispheres. To aid comparison between the empirical and predicted maps, a grid of isopolar angle (solid white) and isoeccentricity (dashed white) lines has been overlaid upon the maps in early visual cortex. The grid was drawn based on the ground truth data and then positioned identically on the predicted maps. Note how

well isopolar angle contours match the predicted maps. A similar quality of fit can be seen for the isoeccentricty lines.

Additionally, we show that our neural network is able to predict nuanced variations in the retinotopic maps across individuals. Figure 2B shows the empirical and predicted polar angle maps for four other participants in the test dataset, which are marked by unusual and/or discontinuous polar angle reversals. In the first three maps (Figure 2B, Participants 2-4), a discontinuous representation of the lower vertical meridian (marked by yellow in the figure) can be observed – indicated by the gray lines. Importantly, these unique variations were correctly predicted by our model. Perhaps even more striking, we see that these borders have merged to form a Y-shape for Participant 5, which was also observed in the predicted map.

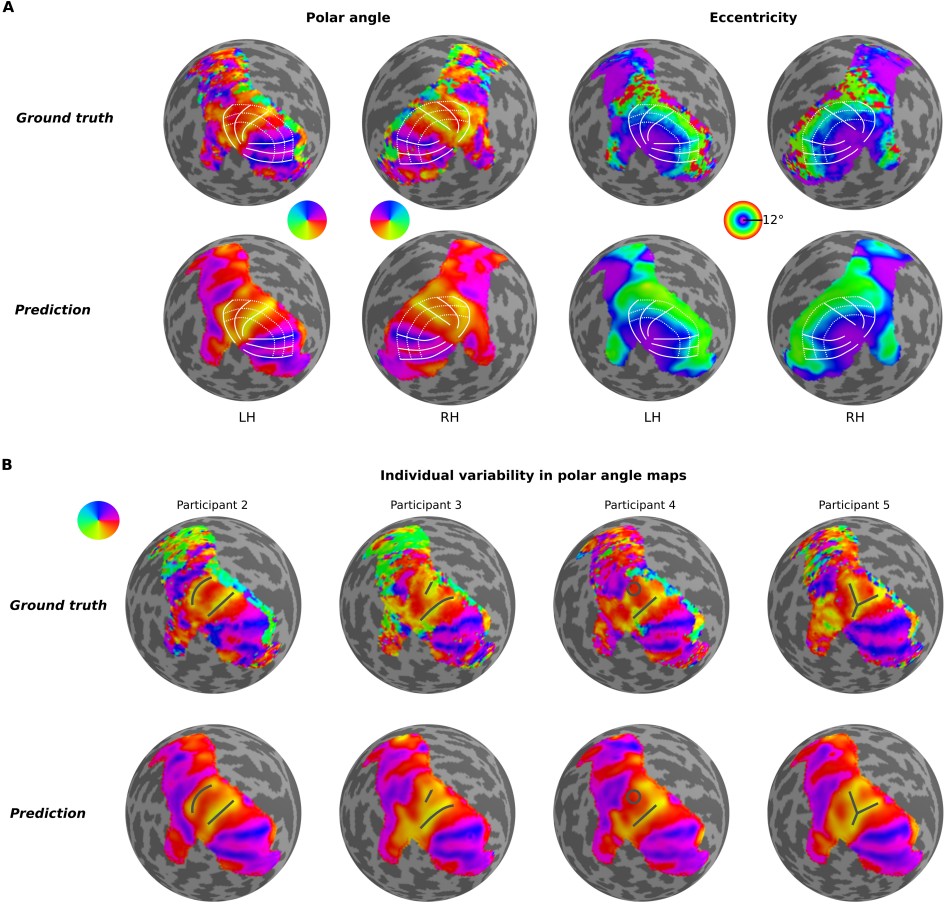

Figure 2: Retinotopy mapping with geometric deep learning.

## 4. Conclusion

Using geometric deep learning and MRI-based neuroimaging data, we were able to predict the detailed functional organization of visual cortex from anatomical features alone. Although we demonstrate its utility for modeling the relationship between brain structure and function in human visual cortex, geometric deep learning is flexible and well-suited for a range of other applications involving data structured in non-Euclidean spaces.

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
