# OpenReview forum: "DeepRetinotopy: Predicting the Functional Organization of Human Visual Cortex from Structural MRI Data using Geometric Deep Learning"
_MIDL.io/2020/Conference — MIDL 2020_

### Official Review · AnonReviewer3 · 2020-03-11
**deep retinotopy**

**Rating:** 2
**Confidence:** 5

**Review:**

This paper applies the geometric deep learning framework by Fey and Lenssen to retinotopic data from 7T fMRI from the HCP dataset.

This is potentially a good application paper. However, there are many limitations. The test results and evaluation are only shown visually and are qualitative. Since the surfaces are all registered, they could simply compute the error between the predicted retinotopic map (continuous scale) and the actual map.

Visually the results look reasonable, although the gray markers that the authors have drawn on the retinotopic maps are distracting. They do draw attention to the similarities between the ground truth and the predicted maps, however, they also ignore the differences. The same goes for the polar angle and the eccentricity maps. In both these cases a numerical evaluation would have been better.

Since the retinotopic maps have considerable inter-subject variability, more insight on the predicted mapping would be valuable.

No information is given about the error evolution and the learning rate. The authors should also provide the mean squared error.

The authors provided curvature and myelin values as an input to the network. Although the loss function only included the retinotopic maps, it would be interesting to see if the loss function incorporated the curvature and myelin (capturing the anatomy), would the retinotopic map prediction be improved. This is because retinotopy is also dependent on the anatomy.

---

### Official Review · AnonReviewer2 · 2020-03-13
**Geometric deep learning applied to structure MRI**

**Rating:** 4
**Confidence:** 3

**Review:**

In the paper, the authors apply geometric deep learning methods to predict the functional organization of the human visual cortex from MRI data. Curvature data, myeling values, and connectivity of vertices on the cortical surface are passed through spline-based convolution layeres to predict the retinotopic map. The network is tested on data from the HCP dataset.

The paper is clearly written, presents an application of geometric deep learning methodology on a medically relevant dataset, with network architecture that seems well-suited for the task. The methodology and application is relevant for the MIDL audience. I believe the paper fits very well the intended focus and scope of a MIDL short paper.

---

### Official Review · AnonReviewer4 · 2020-03-13
**Application of geometric deep learning for the prediction of retinotopic maps.**

**Rating:** 3
**Confidence:** 4

**Review:**

This paper focuses on the application of a geometric deep learning method (spline-based convolutions, Fey et al., 2018) to be able to predict functional retinotopic maps from purely anatomical features using the HCP retinotopy dataset.

Pros:
The paper is written clearly, with the motivation well explained.
The results of this paper are clearly promising, with the method showing not only good prediction of retinotopic maps, but also correctly predicting individual variations.

Cons:
The authors do not provide references or mention whether the prediction of retinotopic maps in the HCP retinotopy dataset has been tried before, therefore it is unknown whether similar results have already been achieved or not.
There is limited description of the method, however, the authors do provide a clear figure of their network.
There is limited validation, and no quantitative results presented in the paper.

---

### Official Review · AnonReviewer1 · 2020-03-14
**Application of geometric deep learning to predict retinotopic mapping**

**Rating:** 3
**Confidence:** 3

**Review:**

This paper is well-written and well-organized. The authors applied geometric deep learning based on spline convolutions to the problem to learn the relationship between the functional organization of visual cortex and anatomy. However, there’s no comparison with other competing methods, so it’s hard to tell whether the proposed method is good enough or not. Also, there’s no quantitative evaluation.

---

### Meta-Review · Area_Chair1 · 2020-04-07
**MetaReview of Paper73 by AreaChair1**

**Rating:** 3

**Metareview:**

The reviewers mostly appear in agreement that this is a well written paper with an interesting application and promising results within the MIDL domain.

**Paper Type:**

validation/application paper

---

### Decision · Program_Chairs · 2020-04-11

Accept